# LEARNING TO PRICE BUNDLES:
# A GCN APPROACH FOR MIXED BUNDLING

## ABSTRACT

Bundle pricing refers to designing several product combinations (i.e., bundles) and determining their prices in order to maximize the expected profit. It is a classic problem in revenue management and arises in many industries, such as e-commerce, tourism, and video games. However, the problem is typically intractable due to the exponential number of candidate bundles. In this paper, we explore the usage of graph convolutional networks (GCNs) in solving the bundle pricing problem. Specifically, we first develop a graph representation of the mixed bundling model (where every possible bundle is assigned with a specific price) and then train a GCN to learn the latent patterns of optimal bundles. Based on the trained GCN, we propose two inference strategies to derive high-quality feasible solutions. A local-search technique is further proposed to improve the solution quality. Numerical experiments validate the effectiveness and efficiency of our proposed GCN-based framework. Using a GCN trained on instances with 10 products, our methods consistently achieve near-optimal solutions (better than 97%) with only a fraction of computational time for problems of small to medium size. It also achieves superior solutions for larger size of problems compared with other heuristic methods such as bundle size pricing (BSP). The method can also provide high quality solutions for instances with more than 30 products even for the challenging cases where product utilities are non-additive.

## 1    INTRODUCTION

Bundle pricing is a widely adopted strategy across industries such as e-commerce, tourism, digital subscriptions, and retail. It refers to the practice that a firm provides combinations (i.e., "*bundles*") of products or services (at discounted prices), supplementing the traditional component pricing (CP) strategy where products are only sold separately, each with its own price. For instance, Amazon Prime combines fast shipping, video streaming, and music services into a single package, and reports show that subscription bundles drive $44 billion in subscription revenue for Amazon in 2024 (Modern Retail, 2024). Similarly, Statista estimates a significant growth of the global digital subscription market, estimated at $832.99 billion in 2023 and projected to reach $1902.28 billion in 2030. This expansion is largely fueled by the increasing adoption of bundled subscription packages, such as Amazon Prime's combination of fast shipping, video streaming, and music services, and Disney+'s bundled offerings with Hulu (Grand View Research, 2023).

In order to optimize this policy, the firm needs to determine which bundles to offer and how to set the corresponding prices, under the constraint that customers will self-select the option that maximizes their surplus. This problem is inherently combinatorial: with $n$ products, the number of possible bundles grows exponentially ($2^n$). Classical formulations, such as the mixed bundling (MB) model by Hanson & Martin (1990), rigorously capture customer surplus constraints but become computationally intractable when $n$ exceeds 15. Later approximations, such as bundle-size pricing (bundles with the same sizes share the same prices), see, e.g., Chu et al. (2011), improve scalability but rely on strong simplifying assumptions (e.g., ignoring product heterogeneity), limiting their realism. Moreover, Chen et al. (2018) show #P-hardness of the bundle pricing problem even under very restricted settings (e.g., size two, discounted bundle-pricing), which further underscores that computational tractability remains the central challenge of bundle pricing in realistic markets.

To overcome this challenge, we propose a GCN-based framework that learns to identify the latent structure of the bundle pricing problem and hence provides high-quality solutions. The core idea of our approach is to train a GCN to make a prediction of which bundle each segment of customer will choose at the optimal solution, which effectively reduces the intractable exponential search space to a much smaller subset. Specifically, we first design a two-layer GCN that learns bundle-segment matchings and predicts likelihoods of each product being in the optimal bundle. The GCN, although trained only on small-scale instances (whose optimal solutions can be efficiently computed), can produce high-quality predictions for large-scale problems. Then, based on these predictions, we develop two strategies to generate candidate solutions. The first is a *Fixed Cutoff Pruning* (FCP) method. In the FCP method, for each segment, we only keep those bundles that contain products with predicted probability greater than 0.5, and discard the rest. The second is a *Progressive Cutoff Pruning* (PCP) method. In the PCP method, for each segment, we rank bundles by their predicted probability and retain the bundles that satisfy a cutoff requirement as candidates to construct prefix bundles for each segment. Based on the pruned space, we apply Hanson & Martin (1990)'s mixed integer linear programming (MILP) formulation to find the optimal prices, which successfully maintain near-optimal revenue performance while saving significant computational costs. To further improve solution quality, we incorporate a local search method to explore alternative promising bundle assortments by strategically adding or dropping products, guided by the likelihood predictions of our GCN model. Finally, we conduct extensive experiments on synthetic datasets with varying product and segment scales, comparing against established baselines including mixed bundling and bundle-size pricing.

Our results demonstrate that for small scale problems (with no more than 10 products), the proposed strategies consistently achieve over 97% of optimal revenue while requiring only a fraction of run-time compared to other baseline methods. Moreover, the integration of local search further improves solution quality, yielding an additional 1% increase in revenue on average within shorter runtime compared to traditional strategies. For medium-scale problems (with 15-25 products), where mixed bundling is intractable, our strategies consistently beats bundle size pricing, in either revenue performance under less than 30 seconds or computational time when achieving similarly effective level of optimality. For large-scale problems (e.g., with more than 30 products) with non-additive utilities, traditional methods become computationally intractable, while our proposed method can still obtain a solution efficiently. To the best of our knowledge, our approach presents the first scalable approach for providing an efficient solution to the bundle pricing problem under the non-additive setting and the numerical experiments demonstrate the effectiveness and scalability of our proposed approach.

## 2 RELATED WORK

The study of bundle pricing has a long history in economics and operations research communities. Early work in the two-product setting established the profitability of bundling relative to separate sales: Stigler (1963) first shows that pure bundling (PB) in which products are only sold together as a package at a single price can dominate component pricing in profitability for two products, while Adams & Yellen (1976) and Schmalensee (1984) demonstrate the profitability for CP, PB and MB for the two-products case. Later research extends these insights to large-scale settings. A stream of work analyzes simplified mechanisms such as CP (Abdallah 2019), PB (Bakos & Brynjolfsson 1999; Abdallah 2019), and bundle-size pricing (BSP) (Chu et al., 2011). Specifically, Bakos & Brynjolfsson (1999) prove that PB approximates the revenue of MB when the number of products grows large under zero costs and independent and additive valuations. Later, Abdallah (2019) provides lower bounds for revenue loss under positive costs from PB. Empirical and theoretical studies show that BSP can outperform CP and PB in certain conditions and can achieve asymptotic optimality when product costs are homogeneous, though it deteriorates under cost heterogeneity (Hitt & Chen, 2005; Chu et al., 2011; Abdallah et al., 2021; Li et al., 2021). More recently, novel mechanisms have been proposed, such as pure bundling with disposal for costs (Ma & Simchi-Levi, 2021), lootbox schemes (Chen et al., 2021), and component pricing with bundle-size discounts (CPBSD) (Chen et al., 2022), which unify CP, PB and BSP approaches and achieve constant-factor guarantees. Complementary to mechanism design, a computational line of work has modeled bundle pricing directly as a mixed-integer program. The seminal work by Hanson & Martin (1990) pioneers the MB formulation of the optimal bundle pricing problem, showing that it preserves rigorous surplus maximization and price sub-additivity constraints, though its scalability is limited by the exponential growth of candidate bundles. Our work follows this computational tradition: we retain Hanson & Martin (1990)'s

framework while leveraging graph-based learning to prune the bundle space before invoking the MILP solver, thereby combining theoretical rigor with practical scalability.

Recently, the use of machine learning to accelerate the solution of combinatorial optimization problems has attracted increasing attention. A seminal work by Gasse et al. (2019) encodes mixed-integer programs as constraint-variable bipartite graphs and trains a GCN to imitate strong branching decisions, accelerating branch-and-bound (B&B) and inspiring a series of follow-up work. For example, Nair et al. (2020) introduce neural diving and neural branching strategies for B&B; Ding et al. (2020) extend the bipartite representation to a tripartite graph to predict partial solutions; Gupta et al. (2022) propose a lookback imitation framework for branching efficiency; Sonnerat et al. (2021) combine neural diving with neural neighborhood search to improve solution quality, and Paulus & Krause (2023) design GCN-based diving methods to facilitate B&B. In the context of linear programming, Fan et al. (2023) propose GCN-based initial basis selection to warm-start simplex and column generation, while Liu et al. (2024) train GCNs to imitate expert pivot policies, shortening pivot paths. These studies confirm that GCNs can substantially reduce computational effort in solving optimization problems. In addition to accelerating optimization solvers, machine learning is also directly integrated into the core structure of operations research problems. Kool et al. (2019) introduce an attention-based model that learns powerful heuristics for complex routing problems like the traveling salesman problem (TSP) and the vehicle routing problem (VRP), achieving near-optimal results. Li et al. (2025) leverage the generalizability of GCNs to solve large-scale NP-hard constrained assortment optimization problems under the multinomial logit choice model. In a similar vein, we apply GCNs directly to the bundle pricing problem, where the GCN learns segment–bundle interactions and predicts promising candidates. In this way, our approach preserves the theoretical rigor of Hanson & Martin (1990)'s model while leveraging GCN predictions to accelerate large-scale instances, thereby extending GCN-assisted optimization to a core revenue management setting.

## 3 PROBLEM DEFINITION AND BASELINES

We study a bundle pricing problem where a seller offers $n$ distinct products to $m$ heterogeneous customer segments. Hence, there are a total of $2^n$ bundle choices, denoted as $\mathcal{B} = \{0, 1, \ldots, L\}$, where $L = 2^n - 1$ and 0 indicates the empty bundle. The seller chooses the set of prices $\{p_b \mid b \in \mathcal{B}\}$ for bundles to maximize profit, which can be defined as $\sum_{k=1}^{m} \sum_{b \in \mathcal{B}} (p_b - c_{kb}) \cdot q_{kb}$, where $c_{kb}$ is the serving cost of selling bundle $b$ to segment $k$ and $q_{kb} \in \{0, 1\}$ denotes whether segment $k$ selects bundle $b$ under self-selection assumption.

The buyers' demand for bundles follows standard assumptions: each segment $k$ has a certain valuation $R_{kb}$ for bundle $b$. After observing the price $p_b$, each buyer would select exactly one bundle that maximizes his/her individual surplus, defined by $s_{kb} = R_{kb} - p_b$, provided that the surplus is non-negative.

Under these assumptions, the seller's revenue optimization problem can be formulated as follows:

$$\max \quad \sum_{k=1}^{m} \sum_{b \in \mathcal{B}} (p_b - c_{kb}) \cdot \alpha_k \, q_{kb} \tag{1}$$

$$\text{s.t.} \quad q_{kb} \leq \mathbb{1}\left( b \in \arg\max_{b' \in \mathcal{B}} \{R_{kb'} - p_{b'}\} \, \wedge \, R_{kb} - p_b \geq 0 \right), \quad \forall k, b \in \mathcal{B} \tag{2}$$

$$\sum_{b \in \mathcal{B}} q_{kb} = 1, \quad \forall k \tag{3}$$

$$p_b \geq 0, \quad \forall b \in \mathcal{B}. \tag{4}$$

Here, $\alpha_k$ is the proportion of buyer segment $k$, and $\mathbb{1}(\cdot)$ is the indicator function that enforces buyers' surplus maximization choice.

In the following discussion, we assume $R_{kb} = f\left( \sum_{j \in b} u_{kj} \right)$ where $u_{kj}$ is the utility of product $j$ for segment $k$, $f(\cdot)$ is an increasing concave function, capturing potential correlations among products.

We assume $c_{kb} = \left( \sum_{j \in b} c_j^u \right) + c_k^s$, where $c_k^s$ is the cost associated with serving customer segment $k$ (e.g., the shipping cost) and $c_j^u$ is the unit cost of product $j$. The concavity of $f(\cdot)$ captures the economic consensus of diminishing marginal utility, and the cost structure captures the principle of economies of scale: since the fixed cost is spread across all items in the bundle, the average cost per item decreases as more products are added. Hanson & Martin (1990) formulated this problem as a mixed-integer linear program (MILP). For the sake of completeness, we describe this formulation in the following.

### 3.1 Mixed Bundling Formulation (Hanson & Martin, 1990)

Consider the problem with all possible bundles denoted by $\mathfrak{B}$. We define the following variables: $\theta_{kb}$ is a binary indicator for whether segment $k$ chooses bundle $b$; $p_b$ is the price of bundle $b$; $P_{kb}$ is the effective price paid by segment $k$; $s_k$ is the surplus of segment $k$; $Z_{kb}$ is the profit from assigning bundle $b$ to segment $k$; and $A_b$ is the set of components which define bundle $b$. The mixed bundle pricing problem can be written as follows:

$$\max \quad \sum_{k=1}^{m} \sum_{b \in \mathfrak{B}} \alpha_k \cdot Z_{kb} \tag{5}$$

$$\text{s.t.} \quad \sum_{b \in \mathfrak{B}} \theta_{kb} = 1, \qquad\qquad \forall k \tag{6}$$

$$s_k \geq R_{kb} - p_b, \qquad\qquad \forall k, b \in \mathfrak{B} \tag{7}$$

$$p_b - R_{\max}(1 - \theta_{kb}) \leq P_{kb}, \qquad\qquad \forall k, b \in \mathfrak{B} \tag{8}$$

$$P_{kb} \leq p_b, \qquad\qquad \forall k, b \in \mathfrak{B} \tag{9}$$

$$s_k = \sum_{b \in \mathfrak{B}} (R_{kb} \theta_{kb} - P_{kb}), \qquad\qquad \forall k \tag{10}$$

$$R_{kb} \theta_{kb} - P_{kb} \geq 0, \qquad\qquad \forall k, b \in \mathfrak{B} \tag{11}$$

$$s_k \geq \sum_{b \in \mathfrak{B}} (R_{kb} \theta_{jb} - P_{jb}), \qquad\qquad \forall k, j \tag{12}$$

$$Z_{kb} = P_{kb} - c_{kb} \theta_{kb}, \qquad\qquad \forall k, b \in \mathfrak{B} \tag{13}$$

$$p_b \leq p_{b_1} + p_{b_2}, \qquad\qquad \text{if } A(b) = A(b_1) \cup A(b_2) \tag{14}$$

$$p_b, P_{kb}, s_k \geq 0, s_{k,0} = 0, \theta_{ki} \in \{0, 1\} \tag{15}$$

We note that we use a simplified price sub-additivity constraints in (13). We refer to Appendix A.3 for the detailed discussions.

In the original formulation in Hanson & Martin (1990), all bundles combinations are included in the formulation, that is $\mathfrak{B} = \mathfrak{F} = 2^{\{1,\dots,n\}}$. We call the corresponding problem with complete combinations HM($\mathfrak{F}$). Meanwhile, one can restrict to a subset of bundles $\mathfrak{B}$ and solve a restricted version of the above problem with only $b \in \mathfrak{B}$. We call the corresponding problem $HM(\mathfrak{B})$.

We note that the main computational challenge of the above formulation lies in the exponential number of variables corresponding to all the possible bundles. In the following, we demonstrate how GCN framework can be leveraged to help solve large-scale bundle pricing problem efficiently.

## 4 GCN-Based Strategies

In this section, we design a GCN-based strategy to solve the optimal bundle pricing problem. Specifically, we adopt a bi-directional GCN architecture, where each directional pass is implemented using a generalized graph convolutional layer that aggregates transformed neighbor and edge features through softmax aggregation (i.e., assigning normalized weights to neighbor messages using the softmax function). Our model predicts a segment-product probability matrix $\mathbb{P} \in \mathbb{R}^{m \times n}$ where $\mathbb{P}_{kj}$ is the predicted likelihood that customer segment $k$ selects product $j$. These probabilities serve as a compact representation of heterogeneous preferences and provide the basis for pruning the exponentially large bundle space. We describe our strategies in detail in the following sections.

## 4.1 GRAPH REPRESENTATION

The figure below is the graph representation of the optimal bundle pricing problem, which serves as the input of the GCN. Two types of nodes are included in the graph representation: product nodes ($\square$), and customer nodes ($\bigcirc$), as shown in Figure 1.

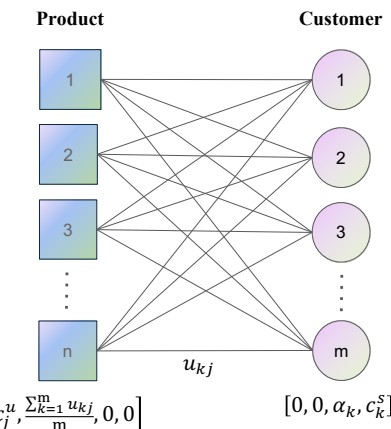

Figure 1: Graph representation of the bundle pricing problem with customer and product nodes.

In Figure 1, $\mathcal{V}$ and $\mathcal{E}$ denote the sets of nodes and edges, respectively. Let $\mathbf{Y}^P$ and $\mathbf{Y}^S$ denote the feature matrices of product nodes and customer nodes, respectively. The feature vector of each product $j$ is $\mathbf{y}_j^P = \begin{bmatrix} c_j^u, & \frac{1}{m}\sum_{k=1}^{m} u_{kj}, & 0, & 0 \end{bmatrix}$; the feature vector of each customer $k$ is $\mathbf{y}_k^S = \begin{bmatrix} 0, & 0, & \alpha_k, & c_k^s \end{bmatrix}$. Let $\mathbf{Z}$ denote the feature matrix of edges. Each edge is characterized by $u_{kj}$, representing the $k$-th customer's utility towards product $j$.

## 4.2 GCN STRUCTURE

We employ two bi-directional message-passing layers based on the Generalized Graph Convolution, with dropout ($p = 0.5$) applied after each fusion. The network will produce edge logits, and probabilities are obtained at inference time by applying a sigmoid function to these logits. We explain the computation details below.

**First bi-directional message-passing layer.** The first message-passing layer in Figure 2 maps the graph in Figure 1 into a new graph with updated node features and the same edge features.

Denote the set of all products by $P$ and the set of all customers by $S$. We then define the forward edge set $\overrightarrow{\mathcal{E}} = \{(j,k) \mid j \in P, \ k \in S\}$, and the backward edge set $\overleftarrow{\mathcal{E}} = \{(k,j) \mid (j,k) \in \overrightarrow{\mathcal{E}}\}$.

The feature matrix $\mathbf{X}$ is formed by stacking the product and customer node features.

$$\mathbf{X} = \begin{bmatrix} (\mathbf{y}_1^P)^\top & \cdots & (\mathbf{y}_n^P)^\top & \mid & (\mathbf{y}_1^S)^\top & \cdots & (\mathbf{y}_m^S)^\top \end{bmatrix}^\top \in \mathbb{R}^{(n+m)\times 4}.$$

For any feature matrix, we let a bold uppercase letter (e.g., $\mathbf{X}$) represent the matrix and the corresponding bold lowercase letter with an index (e.g., $\mathbf{x}_i$) denote its $i$-th row.

Specifically, the bi-directional generalized graph convolutional layer update for each node $i$ in forward (product-to-customer) and backward (customer-to-product) direction is defined as

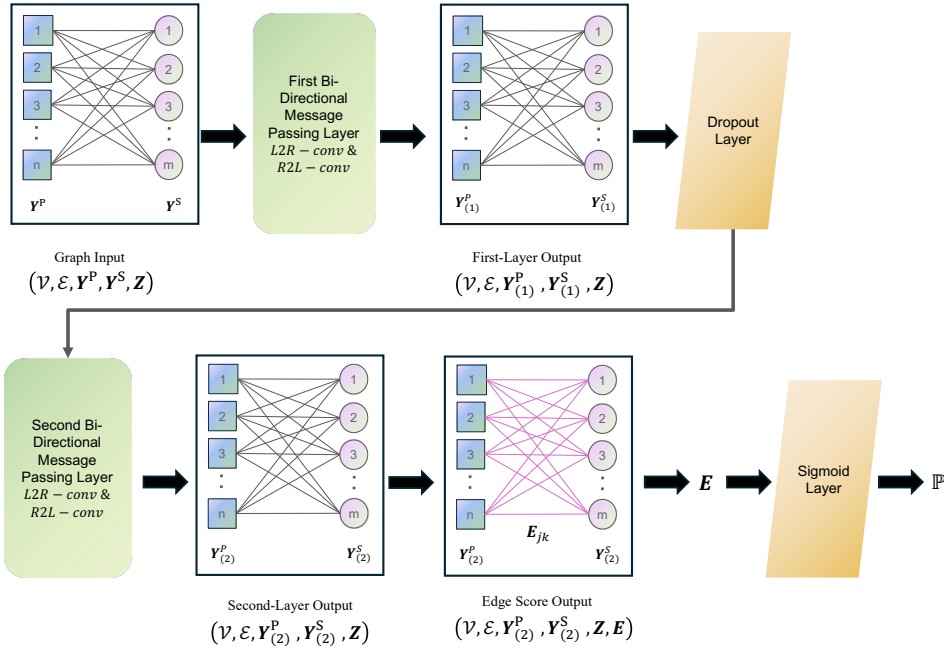

Figure 2: Illustration of GCN network.

$$\mathbf{x}_{i,(1)}^{\text{fw}} = \text{MLP}\left(\mathbf{x}_i + \sum_{j\in\mathcal{N}(i)} \text{Softmax}\big(\phi(\mathbf{x}_j, \mathbf{Z}_{ji})\big) \odot \phi(\mathbf{x}_j, \mathbf{Z}_{ji})\right),$$

$$\mathbf{x}_{i,(1)}^{\text{bw}} = \text{MLP}\left(\mathbf{x}_i + \sum_{j\in\mathcal{N}(i)} \text{Softmax}\big(\phi(\mathbf{x}_j, \mathbf{Z}_{ij})\big) \odot \phi(\mathbf{x}_j, \mathbf{Z}_{ij})\right).$$

Here, $\text{MLP}(\cdot)$ denotes a standard multi-layer perceptron (a sequence of fully connected layers with nonlinear activations). The $\mathcal{N}(i)$ denotes the set of neighbors of node $i$. The operator $\odot$ denotes component-wise multiplication between two vectors (i.e., multiplying elements with the same index), and the softmax is applied component-wise across neighbors for each feature dimension. The message function is defined as $\phi(\mathbf{x}_j, \mathbf{Z}_{ij}) = \text{ReLU}(\mathbf{x}_j + \mathbf{Z}_{ij} \cdot \mathbf{1}) + \epsilon \cdot \mathbf{1}$, where $\epsilon$ is a small constant (we choose $\epsilon$ to be $1 \times 10^{-7}$ in our implementation) for numerical stability. When considering reversed edges, the corresponding edge attributes are also reversed so that each edge feature remains well-aligned.

The node features from both directions are then merged using a mask matrix $\mathbf{M}$ (where diagonal entries are $1$ for product nodes and $0$ otherwise), passed through an entry-wise ReLU activation, and regularized with dropout ($p = 0.5$):

$$\mathbf{X}^{(1)} = \text{Dropout}\big(\text{ReLU}\big((\mathbf{I} - \mathbf{M}) \cdot \mathbf{X}_{(1)}^{\text{fw}} + \mathbf{M} \cdot \mathbf{X}_{(1)}^{\text{bw}}\big)\big).$$

Once the updated feature matrix $X^{(1)}$ is computed, the updated feature matrices for products $Y_{(1)}^P$ and customers $Y_{(1)}^S$ after the first bi-directional layer are obtained by selecting the corresponding rows: $\mathbf{Y}_{(1)}^{\text{P}} = \mathbf{M} \cdot \mathbf{X}_{(1)}, \mathbf{Y}_{(1)}^{\text{S}} = (\mathbf{I} - \mathbf{M}) \cdot \mathbf{X}_{(1)}$.

$\mathbf{X}^{(1)}$ serves as the input for the second layer with different dimensionalities. The input sizes and the output sizes of the first layer are $(n + m, 4)$, and $(n + m, d_{hidden})$, respectively, where we set $d_{hidden} = 128$ in all experiments.

**Second bi-directional message-passing layer.** The construction of the second bi-directional message-passing layer is similar to the first message-passing layer. The only difference is that the input sizes and the output sizes are $(n + m, d_{hidden})$, and $(n + m, d_{hidden})$, respectively.

**Edge score calculation.** After calculating the output $\mathbf{Y}_{(2)}^P$ and $\mathbf{Y}_{(2)}^S$ of the second bi-directional layer, the score $\mathbf{E}_{jk}$ for the edge connecting product $j$ and segment $k$ is calculated as

$$\mathbf{E}_{jk} = (\mathbf{y}_{j,(2)}^P)^\top \mathbf{U}\, \mathbf{y}_{k,(2)}^S + \mathrm{MLP}_{\mathrm{edge}}(\mathbf{Z}_{jk}),$$

where $U \in \mathbb{R}^{d_{hidden} \times d_{hidden}}$ is a learnable bilinear weight matrix, $\mathbf{Z}_{jk}$ is the edge attribute for pair $(j, k)$, and $\mathrm{MLP}_{\mathrm{edge}} : \mathbb{R} \to \mathbb{R}$ is a two-layer feed-forward network (Linear–ReLU–Linear) producing a scalar correction term.

**Output.** The model outputs the logits $\mathbf{E} \in \mathbb{R}^{n \times m}$. In practice, we apply a sigmoid function at inference time to obtain probabilities, $\mathbb{P} = \sigma(\mathbf{E})^\top \in \mathbb{R}^{m \times n}$, where each entry $\mathbb{P}_{kj}$ corresponds to the predicted probability that segment $k$ include product $j$ in his/her purchased bundle.

**Training.** We provide the model training details in Appendix B.

### 4.3 PRUNING-BASED STRATEGIES

In this section, we leverage the probability matrix $\mathbb{P}$ predicted by the GCN model to guide two pruning strategies that construct a compact yet high-quality bundle space. The resulting reduced problem is then solved using the MILP formulation of Hanson & Martin (1990), but restricted to the pruned bundle set rather than the full exponential space.

**Fixed Cutoff Pruning (FCP).** For each segment $k \in [m] = \{1, \ldots, m\}$ and $j \in [n] = \{1, \ldots, n\}$, we define its candidate bundle by a fixed cutoff value. In our default method, we set the fixed cutoff to 0.5. That is $B_k^{\mathrm{FCP}} = \{j \in [n] \mid \mathbb{P}_{kj} \geq 0.5\}$ for $k \in [m]$ and the overall candidate set is $\mathcal{F}^{\mathrm{FCP}} = \{B_k^{\mathrm{FCP}} \mid k \in [m]\}$, with size at most $m$. Here, the overall candidate set is shared across all segments, meaning that any segment is free to select any $B_k^{\mathrm{FCP}}$, not only its own. This reduces the feasible space from $2^n$ to $O(m)$, making the subsequent MILP much more tractable. Then we solve the MILP formulation on the pruned space constructed by FCP (that is, we solve $HM(\mathfrak{F}^{\mathrm{FCP}})$ to obtain the solution).

In the case where all probabilities of a segment fall below the cutoff, we retain the single product with the highest probability to avoid producing an empty bundle.

**Progressive Cutoff Pruning (PCP).** For each segment $k$, we sort products by descending predicted probability and retain those with $\mathbb{P}_{kj} \geq 0.5$. Let $S_k = \big(j_{(1)}, j_{(2)}, \ldots, j_{(|S_k|)}\big)$ denote this ordered set of products. We then construct a sequence of prefix bundles $B_{k,i}^{\mathrm{PCP}} = \{j_{(1)}, j_{(2)}, \ldots, j_{(i)}\}$ for $i = 1, \ldots, |S_k|$. The overall candidate set is defined as the union across all segments: $\mathfrak{F}^{\mathrm{PCP}} = \bigcup_{k=1}^m \{B_{k,i}^{\mathrm{PCP}} \mid i = 1, \ldots, |S_k|\}$, resulting in a size of at most $m \cdot (n + 1)$.

We then solve the MILP formulation on this pruned space (i.e., $HM(\mathfrak{F}^{\mathrm{PCP}})$). Note that for any specific segment $k$, PCP produces a strictly nested chain of bundles:

$$B_{k,1}^{\mathrm{PCP}} \subset B_{k,2}^{\mathrm{PCP}} \subset \cdots \subset B_{k,|S_k|}^{\mathrm{PCP}}.$$

Due to this nested structure, we replace the computationally expensive combinatorial sub-additivity constraints within each segment with a simple chain of monotone price inequalities:

$$p(B_{k,i}^{\mathrm{PCP}}) \leq p(B_{k,i+1}^{\mathrm{PCP}}) \quad \text{for } i = 1, \ldots, |S_k| - 1.$$

Standard sub-additivity constraints are maintained for bundles across different segments.

### 4.4 LOCAL SEARCH STRATEGY

In order to further improve our solution, we propose a local search strategy. The basic idea is to iteratively modify each segment's bundle space by either adding an unselected product or dropping a selected product, and accept modifications if revenue increases.

However, the effectiveness of local search largely depends on an effective search path. Therefore, we develop a preference-based local search strategy guided by the probability matrix predicted by the GCN model. Particularly, when adding a product to the bundle of a segment $k$, we consider the product with the highest $\mathbb{P}_{kj}$ that is unselected; and when dropping a product from the bundle of a segment $k$, we consider the product with the lowest $\mathbb{P}_{kj}$ that is selected.

More precisely, we construct initial bundle assortments by the FCP approach. Here the initial bundles are fixed as segment–bundle assignments, so each segment starts strictly with its own $B_k^{\text{FCP}}$. For each iteration, every segment generates two neighbors: one by adding the highest-probability unselected product and one by dropping the lowest-probability selected product. All neighbors are evaluated sequentially in segment order, and if any improvement is found, the first improving modification is accepted immediately and a new iteration begins from the updated solution (thus a deep-first search is conducted).

The local search process will terminate when a full cycle over all segments completes without any improvement, or when the predefined iteration limit is reached. Furthermore, to save the evaluation cost at each neighbor, we rely on the LP relaxation to evaluate neighbors during iteration. In the LP relaxation, we fix the bundle-segment assignment, and the resulting LP provides a valid lower bound of the optimal value of the MILP (see Appendix A.2 for the LP formulation). Thus, any improvement detected under LP is guaranteed to be valid for MILP, which ensures the accuracy of improvement evaluation. At the same time, LP solves much faster than MILP, greatly reducing the computational cost of each exploration step. We use FCP+LS to denote the corresponding algorithm and provide the detailed pseudo-code in Algorithm 1.

## 5 NUMERICAL EXPERIMENTS

We now present comprehensive numerical experiments designed to evaluate the effectiveness and scalability of our proposed GCN-assisted bundle pricing strategies. The goals of this section are twofold: (i) to demonstrate that our approaches consistently achieve high revenue ratios while maintaining significant computational efficiency, and (ii) to validate their robustness across heterogeneous problem settings with varying numbers of customer segments and products.

To thoroughly test scalability and robustness, we evaluate our strategies under varying numbers of customer segment $m$ and product number $n$. For each scenario, we generate samples using parameters identical to those of the training data, and take the average of 100 samples. All experiments were conducted using the Gurobi Optimizer version 12.0.2 on a machine with an Apple M1 Pro CPU (3.2 GHz) and 16GB RAM, utilizing up to 8 threads.

To evaluate any method, we adopt two normalized metrics: *Revenue Ratio (RR)* and *Time Ratio (TR)*

$$RR_{A,B} = \frac{\text{Revenue of approach A}}{\text{Revenue of approach B}}, \quad TR_{A,B} = \frac{\text{Runtime of approach A}}{\text{Runtime of approach B}}.$$

where RR measures solution quality and TR captures efficiency. These two metrics allow direct comparison of solution quality and speed across different algorithms.

### 5.1 NUMERICAL RESULTS

We compare our proposed approach to two baselines, the mixed bundling (MB) baseline and the bundle size pricing (BSP) baseline:

**Mixed Bundling (MB) baseline:** For MB, we follow Hanson & Martin (1990)'s MILP formulation. The MB formulation is a MILP that assigns each bundle with its own price respectively. Detailed formulation is provided in Section 3.1.

**Bundle Size Pricing (BSP) baseline:** The BSP baseline is proposed by Chu et al. (2011). The BSP approximates MB by assigning the same price to all bundles of equal size. Detailed formulation is provided in Appendix A.1.

In Table 5.1, we report the numerical results of our algorithms compared with both baselines, for problem with $n = 10$ and varying number of segments. From Table 5.1, we can see that our three strategies all maintain more than 97.5% of the optimal revenue while only requiring a fraction of the

computation time of the MB baseline. Meanwhile, the BSP approach often has a significant profit loss compared to the MB baseline. Among the strategies we propose, we find that the PCP strategy achieves around 1% higher revenue than FCP by retaining a larger candidate space, while FCP+LS also gives a 1% revenue improvement over the plain FCP approach.

Table 5.1: Performance of GCN-based strategies vs. Mixed Bundling baseline ($n = 10$).

| **Strategy** | $m = 10, n = 10$ | | | $m = 20, n = 10$ | | | $m = 30, n = 10$ | | |
|---|---|---|---|---|---|---|---|---|---|
| | $RR_{\cdot,MB}$ | Time (s) | $TR_{\cdot,MB}$ | $RR_{\cdot,MB}$ | Time (s) | $TR_{\cdot,MB}$ | $RR_{\cdot,MB}$ | Time (s) | $TR_{\cdot,MB}$ |
| FCP | 0.9836 | 0.0558 | 0.0106 | 0.9785 | 0.3767 | 0.0130 | 0.9757 | 1.2276 | 0.0151 |
| PCP | 0.9907 | 0.7587 | 0.1383 | 0.9874 | 9.0669 | 0.2886 | 0.9862 | 71.5509 | 0.6835 |
| FCP+Local Search | 0.9945 | 0.9700 | 0.1799 | 0.9877 | 7.0461 | 0.2410 | 0.9850 | 30.1076 | 0.4158 |
| BSP | 0.8990 | 0.0382 | 0.0070 | 0.8796 | 0.1861 | 0.0064 | 0.8669 | 0.5386 | 0.0014 |

For problems with more than 10 products, calculating the optimal solution of the MILP model under the mixed bundling baseline is computationally challenging. Therefore, we use the BSP as baseline to test problems with larger than 10 products. The results are reported in Table 5.2. From Table 5.2, we can see that the plain FCP approach can achieve comparable performance with the BSP approach with a fraction of computation time, while the PCP approach can achieve significantly better solution than the BSP approach with more computational time (the total time is still less than 30 seconds).

Table 5.2: Comparison of different strategies across various problem sizes.

| | **FCP** | | | **PCP** | | | **FCP+Local Search** | | |
|---|---|---|---|---|---|---|---|---|---|
| Scenario | $RR_{\cdot,BSP}$ | Time (s) | $TR_{\cdot,BSP}$ | $RR_{\cdot,BSP}$ | Time (s) | $TR_{\cdot,BSP}$ | $RR_{\cdot,BSP}$ | Time (s) | $TR_{\cdot,BSP}$ |
| $m = 10, n = 15$ | 1.060 | 0.043 | 0.380 | 1.150 | 2.056 | 17.975 | 1.136 | 0.891 | 8.012 |
| $m = 15, n = 15$ | 1.074 | 0.125 | 0.588 | 1.167 | 9.921 | 47.579 | 1.150 | 3.987 | 19.278 |
| $m = 20, n = 15$ | 1.082 | 0.307 | 0.829 | 1.171 | 27.215 | 71.445 | 1.156 | 13.884 | 37.170 |
| $m = 10, n = 20$ | 1.007 | 0.044 | 0.331 | 1.147 | 4.799 | 36.676 | 1.116 | 1.150 | 8.665 |
| $m = 10, n = 25$ | 0.953 | 0.048 | 0.243 | 1.146 | 8.250 | 41.401 | 1.092 | 1.360 | 6.945 |

**Scalability with product size.** Table 5.3 illustrates how FCP and PCP scale with the number of products $n$ when the number of segments is fixed at 10. A key observation is that FCP maintains a nearly constant candidate set size $O(m)$, resulting in stable runtimes around 0.03 seconds regardless of $n$. By contrast, PCP exhibits $O(n)$ growth in candidate space. And both methods enable tractable solutions of bundle pricing instances up to $n = 100$, which would be intractable under exact mixed bundling. This highlights our ability to combine strong pruning with broad scalability across large problem sizes.

Table 5.3: Scalability of FCP and PCP strategies across different product sizes $n$ ($m = 10$).

| $n$ | FCP Time (s) | PCP Time (s) | $RR_{FCP,PCP}$ | FCP Bundles | PCP Bundles |
|---|---|---|---|---|---|
| 10 | 0.107 | 0.843 | 0.972 | 10 | 61 |
| 20 | 0.032 | 5.440 | 0.999 | 10 | 137 |
| 30 | 0.034 | 21.103 | 0.990 | 10 | 220 |
| 40 | 0.032 | 36.134 | 0.982 | 10 | 282 |
| 50 | 0.034 | 59.050 | 0.968 | 10 | 359 |
| 60 | 0.026 | 73.665 | 0.951 | 10 | 431 |
| 70 | 0.030 | 122.706 | 0.910 | 10 | 503 |
| 80 | 0.028 | 184.814 | 0.897 | 10 | 572 |
| 90 | 0.029 | 206.681 | 0.880 | 10 | 647 |
| 100 | 0.030 | 283.578 | 0.852 | 10 | 710 |

**Scalability with customer size.** Table 5.4 evaluates the scalability of our proposed methods as $M$ increases from 10 to 90. While PCP theoretically explores a more comprehensive bundle space, the empirical results indicate that its marginal revenue advantage diminishes at larger customer sizes.

In contrast, FCP captures the same revenue as PCP in just 2.3 seconds. This demonstrates that for large $M$, the heavy computation of PCP is unnecessary: it runs hundreds of times slower than FCP without providing any additional profit.

Table 5.4: Scalability of FCP and PCP strategies across different parameter $M$ ($N = 25$).

| $M$ | FCP Time (s) | PCP Time (s) | $RR_{FCP,PCP}$ | FCP Bundles | PCP Bundles |
|---|---|---|---|---|---|
| 10 | 0.033 | 2.189 | 0.998 | 10 | 135 |
| 30 | 0.202 | 73.234 | 0.997 | 30 | 384 |
| 50 | 0.548 | 485.553 | 0.997 | 50 | 648 |
| 70 | 1.115 | 849.410 | 1.000 | 70 | 886 |
| 90 | 2.256 | 1204.020 | 1.000 | 90 | 1125 |

## 6 CONCLUSION

This paper introduces a learning-guided framework for solving the bundle pricing problem. We leverage GCNs to learn segment–product preference structures under the non-additive setting, and use these predictions to prune the exponential candidate bundle space. The pruned feasible region is then solved with Hanson & Martin (1990)'s MILP formulation, thereby retaining the rigor of product heterogeneity while substantially extending the tractable problem size. Coupled with a probability-guided DFS local search, our framework demonstrates robustness and scalability across different problem sizes. Numerical experiments show that our approach provides solutions that are near-optimal at tractable scales, and scalable to much larger settings while outperforming other heuristics. Therefore, our work provides a near-optimal and efficient solution for large-scale bundle pricing problem.

## 7 CODE OF ETHICS STATEMENT

The authors of this work adhere to the ICLR Code of Ethics. This research on bundle pricing optimization was conducted with the principles of academic integrity and rigor. We have considered the potential societal impacts of our work. While any advanced pricing algorithm could potentially be used for exploitative price discrimination, the primary goal of this research is to improve economic efficiency, which can lead to better value propositions for consumers and more sustainable business models for firms. We believe the societal benefits of more efficient market mechanisms outweigh the potential risks, which can be mitigated by fair business practices and regulation. The experiments in this paper were conducted on synthetic datasets and do not involve sensitive or personally identifiable information.

## 8 REPRODUCIBILITY STATEMENT

To ensure the reproducibility of our research, we provide a comprehensive suite of materials. The complete source code for our proposed GCN-based policies (FCP, PCP, and FCP+LS), baseline implementations, and all numerical experiments is available in the supplementary materials. The mathematical formulations for the baselines are detailed in 3.1 and Appendix A.1. Key hyperparameters and the versions of the main software libraries used (e.g., PyTorch, PyTorch Geometric, Gurobi) are also documented in the Appendix E to facilitate the replication of our results.

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

## A FORMULATIONS FOR BUNDLE PRICING

### A.1 BUNDLE SIZE PRICING (BSP) FORMULATION CHU ET AL. (2011)

**Additional Variables:**

- $p_s$: price for bundles of size $s \in \{0, \dots, n\}$;
- $P_{ks} = p_s \theta_{ks}$.

**Objective:**

$$\max \quad \sum_{k=1}^{m} \sum_{s=0}^{n} \alpha_k \cdot Z_{ks} \tag{16}$$

**Constraints:**

$$s_k \geq R_{ks} - p_s, \quad \forall k, s \tag{17}$$

$$\sum_{s=0}^{n} \theta_{ks} = 1, \quad \forall k \tag{18}$$

$$P_{ks} \leq p_s, \quad \forall k, s \tag{19}$$

$$P_{ks} \geq p_s - M(1 - \theta_{ks}), \quad \forall k, s \tag{20}$$

$$Z_{ks} = P_{ks} - c_{ks}\theta_{ks}, \quad \forall k, s \tag{21}$$

$$s_k = \sum_{s=0}^{n} (R_{ks}\theta_{ks} - P_{ks}), \quad \forall k \tag{22}$$

$$s_k \geq \sum_{s=0}^{n} (R_{ks}\theta_{js} - P_{js}), \quad \forall k, j \neq k \tag{23}$$

$$R_{ks}\theta_{ks} - P_{ks} \geq 0, \quad \forall k, s \tag{24}$$

$$p_{s_1+s_2} \leq p_{s_1} + p_{s_2}, \quad \forall s_1, s_2 \text{ s.t. } s_1 + s_2 \leq n \tag{25}$$

$$p_{s+1} \geq p_s, \quad \forall s = 0, \dots, n-1 \tag{26}$$

$$p_s, P_{ks}, Z_{ks} \geq 0, \quad \theta_{ks} \in \{0, 1\} \quad \forall k, s$$

### A.2 LP FORMULATION (FIXED ASSIGNMENT)

**Decision variables:**

- $p_i \geq 0$: price of bundle $i \in F$, where $F$ is the pruned candidate bundle set;

- $s_k$: surplus of segment $k \in \{1, \ldots, m\}$.

**Objective:**

$$\max_{p,s} \quad \sum_{k=1}^{m} \alpha_k \cdot \left( p_{b_k} - c_{k,b_k} \right), \tag{L.1}$$

where $b_k \in F$ is the fixed bundle assigned to segment $k$.

**Constraints:**

$$
\begin{aligned}
s_k &\geq R_{k,i} - p_i \quad \forall k = 1, \ldots, m, \ \forall i \in F & \text{(lower bound)} & \tag{L.2} \\
s_k &\leq R_{k,b_k} - p_{b_k} \quad \forall k & \text{(assignment binding)} & \tag{L.3} \\
p_i &\leq p_{j_1} + p_{j_2}, \quad \text{if } A_i = A_{j_1} \cup A_{j_2}, \ \{j_1, j_2\} \subset F & \text{(sub-additivity)} & \tag{L.4} \\
p_0 &= 0 & \text{(empty bundle price)} & \tag{L.5}
\end{aligned}
$$

*Remarks.*

- Bundle assignments are fixed externally by assigning each segment with their exact FCP optimal bundle prediction: $\theta_{kb_k} = 1$ and $\theta_{ki} = 0$ for $i \neq b_k$.

- Constraints (L.2) and (L.3) together ensure incentive compatibility: $s_k = R_{k,b_k} - p_{b_k} = \max_{i \in F}\{R_{k,i} - p_i\}$.

- Only bundles in $F$ (predicted or assigned) have price variables, reducing dimensionality.

- The model is a pure LP without integer decision variables, in contrast to the MILP formulations.

- The optimal value of the LP formulation is a lower bound of the corresponding MILP with possible bundle set $\mathfrak{F}$. That is, the optimal value of this LP is less than or equal to that of $HM(\mathfrak{F})$.

### A.3 MODIFIED PRICE SUB-ADDITIVITY CONSTRAINTS

Our mixed bundling follows Hanson & Martin (1990)'s formulation in terms of decision variables, objective, and most constraints (consumer surplus, single price schedule, etc.). The only difference lies in the treatment of price sub-additivity. To enforce the general K-way Cover sub-additivity ($S_{C,K}$) proposed by Hanson & Martin (1990), we introduce a more parsimonious set of constraints. Specifically, we only enforce sub-additivity on 2-way partitions ($S_{P,2}$), which significantly reduces the number of price sub-additivity constraints while being sufficient to guarantee the general condition. The equivalence between these two are proved in Appendix C.

**Definition 1** (K-way Cover sub-additivity — Statement $S_{C,K}$). *For any bundle $B$ and any (potentially overlapping) **cover** by $K \geq 2$ sub-bundles $\{B_1, \ldots, B_K\}$, the following holds:*

$$p(B) \leq \sum_{j=1}^{K} p(B_j).$$

**Definition 2** (2-way Partition sub-additivity — Statement $S_{P,2}$). *For any bundle $B$ and any **disjoint** partition into two sub-bundles $\{B_1, B_2\}$, the following holds:*

$$p(B) \leq p(B_1) + p(B_2).$$

## B MODEL TRAINING AND DATA GENERATION

In this section, we provide a detailed description of the data generation process, the generation of ground-truth labels using an exact solver, and the specific hyperparameters used to train our Graph Convolutional Network (GCN).

## B.1 SYNTHETIC DATA GENERATION

To train our model, we generated a dataset of 5,000 synthetic instances. Each instance represents a mixed bundling problem with $n = 10$ products and $m = 10$ customer segments. The data generation process follows the economic assumptions outlined in Section 3, specifically utilizing a concave utility function $f(\cdot) = \sqrt{\cdot}$ to model diminishing marginal returns. The generation procedure is detailed below:

- **Costs** ($c_j^u, c_k^s$): The unit cost for each product $c_j^u$ is sampled from a uniform distribution $U(0, 1)$. The shipping/serving cost for each segment $c_k^s$ is also sampled from $U(0, 1)$. To ensure profitability is non-trivial, the total cost is scaled by a factor of 0.2.

- **Base Utilities** ($u_{kj}$): The base utility of product $j$ for customer segment $k$, denoted as $u_{kj}$, is sampled from $U(0, 1)$.

- **Reservation Prices** ($R_{kb}$): We assume a non-additive valuation structure. The reservation price of segment $k$ for a bundle $b$ is calculated using a square-root function:

$$R_{kb} = \sqrt{\sum_{j \in b} u_{kj}} \tag{27}$$

- **Segment Sizes** ($\alpha_k$): We generate raw size weights $x_k \sim U(0, 1)$ and normalize them such that the segment proportions sum to 1, i.e., $\alpha_k = x_k / \sum_i x_i$.

## B.2 LABEL GENERATION VIA MIXED BUNDLING

For each of the 5,000 synthetic instances, we obtained ground-truth optimal bundle assignments by solving the Mixed Bundling formulation (Hanson & Martin, 1990).

We set a time limit of 600 seconds and a MIP gap tolerance of $1\%$ for the solver. The solver outputs the optimal binary assignment variables $\theta_{kb}^*$. These are converted into edge labels for the GCN training:

$$y_{kj} = \begin{cases} 1 & \text{if product } j \text{ is included in the optimal bundle for segment } k \\ 0 & \text{otherwise} \end{cases} \tag{28}$$

This results in a supervision signal for every edge in the bipartite graph. Instances where the solver could not find a feasible solution within the time limit were excluded from the dataset.

## B.3 TRAINING PROTOCOL

The 5000 instances were split into a training set (80%) and a validation set (20%). The model we use trains for 500 epochs with batch size 512, learning rate 0.01, 128 hidden channels, bilinear edge scoring, 0.5 dropout, $\epsilon = 1 \times 10^{-7}$, Adam optimizer, and early stopping patience of 50 epochs.

**Loss Function:** To train the model, we formulate the edge prediction task as a binary classification problem. We utilize Binary Cross Entropy with Logits Loss. The loss $\mathcal{L}$ is calculated based on two primary components:

1. **Predicted Logits** ($\mathbf{E}_{kj}$): The raw edge scores output by the GCN bilinear head for the edge connecting customer segment $k$ and product $j$.

2. **Ground Truth Labels** ($y_{kj}$): A binary label derived from the exact solver, where $y_{kj} = 1$ if product $j$ belongs to the optimal bundle for segment $k$, and $y_{kj} = 0$ otherwise.

To address the inherent class imbalance (as optimal bundles typically contain only a small subset of available products), we introduce a positive weight parameter $w_{pos}$. The loss function for a single edge is defined as:

$$\mathcal{L}(\mathbf{E}_{kj}, y_{kj}) = - [w_{pos} \cdot y_{kj} \cdot \log(\sigma(\mathbf{E}_{kj})) + (1 - y_{kj}) \cdot \log(1 - \sigma(\mathbf{E}_{kj}))] \tag{29}$$

where $\sigma(\cdot)$ is the sigmoid function. The weight $w_{pos}$ is calculated dynamically from the training data as $w_{pos} = (1 - p)/p$, where $p$ is the proportion of positive edges across the entire training set.

# C PROOF OF EQUIVALENCE BETWEEN PRICE SUB-ADDITIVITY CONSTRAINTS

## C.1 DEFINITIONS

Let $\mathcal{U}$ be the set containing all products, and let $p : \mathcal{P}(\mathcal{U}) \to \mathbb{R}_{\geq 0}$ be the price function, where $\mathcal{P}(\mathcal{U})$ denotes the power set of $\mathcal{U}$.

**Definition 3** (Price Monotonicity — Statement $\mathcal{M}$). *For any two bundles $A$ and $B$, if $A \subseteq B$, then $p(A) \leq p(B)$.*

**Definition 4** (K-way Partition sub-additivity — Statement $S_{P,K}$). *For any bundle $B$ and any **disjoint** partition into $K \geq 2$ sub-bundles $\{B_1, \ldots, B_K\}$, the following holds:*

$$p(B) \leq \sum_{j=1}^{K} p(B_j).$$

## C.2 THEOREM

**Proposition 1.** *Statement $S_{P,2}$ and Statement $S_{P,K}$ are equivalent.*

*Proof.* We prove the proposition in both directions.

*Direction 1: $S_{P,K} \implies S_{P,2}$* This is true by definition. $S_{P,2}$ is a special case of $S_{P,K}$ where we choose the number of partitions $K = 2$.

*Direction 2: $S_{P,2} \implies S_{P,K}$* We use mathematical induction on the number of partitions, $k$. The base case ($k = 2$) is Statement $S_{P,2}$, which is assumed to be true. Assume the statement holds for $k$ partitions (Inductive Hypothesis). We prove for $K = k + 1$. Let $B$ be partitioned into $\{B_1, \ldots, B_{k+1}\}$. Let $B' = \bigcup_{j=1}^{k} B_j$. Then $B = B' \cup B_{k+1}$ is a 2-way disjoint partition. By $S_{P,2}$:

$$p(B) \leq p(B') + p(B_{k+1}). \tag{30}$$

By the inductive hypothesis, $p(B') \leq \sum_{j=1}^{k} p(B_j)$. Substituting this yields:

$$p(B) \leq \sum_{j=1}^{k+1} p(B_j).$$

Thus, $S_{P,2} \implies S_{P,K}$. $\qquad\square$

**Proposition 2.** *Assuming Price Monotonicity ($\mathcal{M}$), Statement $S_{P,K}$ and Statement $S_{C,K}$ are equivalent.*

*Proof.* We prove the proposition in both directions, assuming $\mathcal{M}$ holds.

*Direction 1: $S_{C,K} \implies S_{P,K}$.* A disjoint partition is a special case of a cover. Thus, if the rule holds for any cover, it must hold for any disjoint partition.

*Direction 2: $(S_{P,K} \wedge \mathcal{M}) \implies S_{C,K}$.* We show that $(S_{P,2} \wedge \mathcal{M}) \implies S_{C,2}$ (the binary case). Let $B = B_1 \cup B_2$ be a cover. We can write $B$ as a disjoint partition: $B = (B_1 \setminus B_2) \cup B_2$. By $S_{P,2}$:

$$p(B_1 \cup B_2) \leq p(B_1 \setminus B_2) + p(B_2). \tag{31}$$

Since $(B_1 \setminus B_2) \subseteq B_1$, by Monotonicity ($\mathcal{M}$), we have $p(B_1 \setminus B_2) \leq p(B_1)$. Substituting this gives $p(B_1 \cup B_2) \leq p(B_1) + p(B_2)$, which proves $S_{C,2}$.

Since $S_{P,K} \iff S_{P,2}$ (from Prop. 1) and $S_{C,K} \iff S_{C,2}$ (by induction), it follows that $(S_{P,K} \wedge \mathcal{M}) \implies S_{C,K}$. $\qquad\square$

**Theorem 1.** *Assuming Price Monotonicity ($\mathcal{M}$), the 2-way Partition sub-additivity ($S_{P,2}$) is equivalent to the K-way Cover sub-additivity ($S_{C,K}$).*

*Proof.* From Proposition 1, $S_{P,2} \iff S_{P,K}$, and from Proposition 2 under Price Monotonicity $\mathcal{M}$, $S_{P,K} \iff S_{C,K}$. Hence $S_{P,2} \iff S_{C,K}$. $\qquad\square$

# D PSEUDOCODE FOR THE LOCAL SEARCH STRATEGIES

---

**Algorithm 1** Segment-wise Local Search

---

1: **Input:** Initial solution $x$, probability matrix $\mathbb{P}$, max_iter
2: **Output:** Optimized solution $x^*$
3: $x^* \leftarrow$ MILP-Init($\{\mathcal{B}_k\}$), $rev^* \leftarrow$ LP-Eval($x^*$)
4: $iter \leftarrow 0$
5: **while** $iter <$ max_iter **do**
6:     $iter \leftarrow iter + 1$, $improve \leftarrow$ FALSE
7:     **for** $seg = 1$ to $m$ **do**
8:         Generate two neighbors of $x^*$ in segment $seg$:
9:             (i) Add the highest-probability unselected product
10:             (ii) Drop the lowest-probability selected product
11:         **for** neighbor $y$ (in the above order) **do**
12:             $(feas, rev) \leftarrow$ LP-Eval($y$)
13:             **if** $feas$ and $rev > rev^* + \epsilon$ **then**
14:                 $x^* \leftarrow y$, $rev^* \leftarrow rev$
15:                 $improve \leftarrow$ TRUE
16:                 **break and restart next iteration**                    ▷ greedy accept
17:             **end if**
18:         **end for**
19:         **if** $improve$ **then**
20:             **break**                    ▷ restart from next iteration
21:         **end if**
22:     **end for**
23:     **if** not $improve$ **then**
24:         **break**                    ▷ full cycle, no improvement
25:     **end if**
26: **end while**
27: **return** $x^*$

---

## E  SOFTWARE VERSION

All experiments are conducted using PyTorch 2.5.1 with PyTorch Geometric 2.6.1 and Gurobi 12.0.2, without CUDA support. The model we use trains for 500 epochs with batch size 512, learning rate 0.01, 128 hidden channels, bilinear edge scoring, 0.5 dropout, $\epsilon = 1 \times 10^{-7}$, Adam optimizer, and early stopping patience of 50 epochs.

## F  LLM USAGE

The authors acknowledge the use of a large language model (LLM) in preparing this paper. Its use was limited to assisting with language polishing, such as improving grammar and clarity, and helping to generate and debug code snippets for the numerical experiments. All final text and code were reviewed and validated by the authors, who take full responsibility for the content of this work.

