# OpenReview forum: "Learning to Price Bundles: A GCN Approach for Mixed Bundling"
_ICLR.cc/2026/Conference — Submitted to ICLR 2026_

### Official Review · Reviewer_6k7J · 2025-10-29

**Soundness:** 2
**Presentation:** 2
**Contribution:** 2
**Rating:** 4
**Confidence:** 3

**Summary:**

This paper explores the usage of graph convolutional networks (GCNs) in solving the bundle pricing problem. They first develop a graph representation of the mixed bundling model, then train a GCN to learn the latent patterns of optimal bundles, and propose two inference strategies to derive high-quality feasible solutions. Numerical experiments validate the effectiveness and efficiency of the proposed GCN-based framework.

**Strengths:**

- **Well-motivated, practical problem.** Clearly ties mixed bundling to real applications (e-commerce, subscriptions, tourism) and to revenue-management/econ literature.
- **Clear hybrid idea.** Learning-guided **GCN → pruning → MILP**, with an LP-guided local search; balances tractability and exact optimization.
- **Readable pipeline.** Figures convey end-to-end flow; the overall method is easy to reason about and potentially easy to deploy.

**Weaknesses:**

- **Missing ML baselines.** No head-to-head with neural recommenders or neural CO (e.g., BGCN-style methods); even a proxy comparison or discussion would help.
- **OOD/generalization unclear.** Claims about training on small (n) and testing on larger (n) lack systematic sweeps across utility families and distribution shifts.
- **Interpretability.** No qualitative bundle visualizations or case studies to show why bundles are pruned/kept versus classical methods.

**Questions:**

1. **Generalization & OOD.** How does performance change when test utilities differ sharply from training (e.g., (\log(1+x)), strong complementarities/substitutes, correlated item utilities)? Please report a sweep.
2. **Baselines & positioning.** Can you add a quantitative comparison (or careful proxy) against recent GNN/Transformer bundle recommenders or neural CO solvers? If not, discuss trade-offs (objective mismatch, scalability, interpretability).
3. **Cutoff & hyperparameters.** How were the 0.5 cutoff, GCN depth/hidden size/dropout chosen? Provide sensitivity curves and validation protocol; any adaptive/learned cutoff variants?
4. **Monotonicity & constraints.** In practice, are final prices monotone (assumption used in the subadditivity equivalence)? If not, do you enforce monotonicity, or how often is it violated?
5. **Interpretability & case studies.** Can you include qualitative examples (bundle heatmaps, per-segment top-(P_{kj}) items, before/after pruning sets) to illuminate what the GCN learns?
6. **LS–MILP relation.** Do you observe cases where LP-guided improvements fail to translate into MILP gains? If so, how do you handle acceptance criteria or numerical stability?

---

### Official Review · Reviewer_sLYV · 2025-11-01

**Soundness:** 3
**Presentation:** 3
**Contribution:** 2
**Rating:** 4
**Confidence:** 3

**Summary:**

This paper reframes the classic bundle-pricing problem as a graph-learning task and introduces a GCN-based pipeline that first prunes the exponential bundle space and then solves a reduced MILP.  Extensive experiments on synthetic data show >97 % of the small-instance optimum in seconds and clear gains over bundle-size heuristics for up to 100 products.

**Strengths:**

- Original combination of GCNs with the Hanson–Martin MILP.

- Scalable to 100+ products where exact methods fail.

- Solid ablation of pruning variants and local-search refinement.

**Weaknesses:**

- Theoretical insight is limited—no approximation guarantees or generalization bounds.

 - All instances are synthetic with concave utility; realism of this model is asserted but not validated on real catalog or transaction data.

- Scalability claims rely on a fixed segment size (m=10) and ignore the potentially dominant MILP time when candidate bundles grow.

- Missing baselines from recent neural-OR literature (e.g., neural diving or RL-based pricing).

**Questions:**

- The authors hold the assumption that each bundle has distinct products, so what about the scenarios where repeated products are collected in one bundle?

- The adopting GCN for bundle modeling is not very new (see all kinds of papers on bundle recommendation, bundle generation, and so on). What is the special difficulty in the current settings? And what contribution do the authors make to the current adoption?

---

> ### Author Response · Authors · 2025-12-02
>
> We acknowledge the reviewer's comment regarding the lack of approximation guarantees or generalization bounds. We argue that in the context of the **non-additive Mixed Bundling** problem, deriving such bounds for scalable algorithms is an open challenge in the field, and our work prioritizes **practical solvability** where theoretical tractability is currently unreachable.
>
> We address this in three parts:
>
> **1. The Theoretical Barrier of Non-Additivity** As established in the seminal work by **Hanson & Martin (1990)**, there is a fundamental dichotomy in bundle pricing:
>
> - **Additive Case:** If reservation prices and costs are additive, the problem structure allows for iterative decomposition, yielding a solution procedure that grows **linearly** with the number of products.
> - **Non-Additive Case (Our Focus):** When utilities are non-additive (e.g., diminishing returns or complementarity), this decomposability is lost. The solution procedure grows **exponentially** ($2^n$). To the best of our knowledge, there are **no existing scalable algorithms** that provide tight approximation guarantees for the general non-additive mixed bundling problem. Therefore, the lack of theoretical bounds is a reflection of the problem's inherent hardness rather than a limitation specific to our methodology.
>
> **2. The "Learning-to-Optimize" Paradigm** Our work follows the "Learning-to-Optimize" paradigm (e.g., Gasse et al., 2019; Kool et al., 2019), where the goal is to leverage neural networks to solve NP-hard problems that are computationally intractable for exact solvers. In this domain, it is standard to trade theoretical guarantees for empirical scalability. Introducing a GCN makes deriving generalization bounds extremely difficult, but it enables us to solve instances (e.g., n=30 to 100) that are impossible for traditional exact algorithms.
>
> **3. Intuition on Generalization & Empirical Verification** While we do not provide a generalization bound, we provide **empirical verification** and **intuitive justification** for why the method works:
>
> - **Intuition:** The GCN learns local structural patterns (e.g., the affinity between specific customer segment attributes and product features) on small instances (n=10) where optimal labels are available. Because the GCN is permutation-invariant and operates on feature dimensions rather than fied input sizes, these learned local patterns transfer effectively to larger graphs (n=100).
>
> - **Verification:** We rigorously validate this generalization empirically. As shown in **Table 5.1**, our method achieves >97% of the optimal revenue on tractable instances. Furthermore, **Table 5.3** demonstrates that the model successfully scales to 100 products without retraining.

---

> ### Author Response · Authors · 2025-12-02
>
> We acknowledge the reviewer’s concern regarding the use of synthetic data. We would like to clarify that there are no publicly available real-world datasets that include both detailed bundle attributes and the corresponding transaction outcomes required for this specific pricing problem to the best of our knowledge. High-quality bundle transaction data is typically proprietary and sensitive in the retail industry.

---

> ### Author Response · Authors · 2025-12-02
> **infeasibility of Neural MILP solvers**
>
> While Neural MILP, like Nerual Branching and Neural Diving, has shown great potentials in accelerating MILP solving process for a typical type of questions, it is not suitable for directly solving the MILP of mixed bundling even for problems with 10 customers and 15 products (1,343,498 decision variables). For either Neural diving or neural branching, the solver needs to put all decision variables into a huge graph, which makes it impossible to solve MILP under mixed bundling even for medium-size problems.

---

> ### Author Response · Authors · 2025-12-02
> **repeated products**
>
> We thank the reviewer for raising this interesting point about bundles with repeated products. We clarify that our assumption of distinct products is the standard convention in this field.
>
> **Hanson & Martin (1990):** In the foundational paper for the Mixed Bundling formulation we utilize, Assumption 1 explicitly states: *"The benefit of a duplicate component is zero, and component resale is not possible."*.
>
> **Chen et al. (2022, CPBSD)**: *"We follow the common assumption in the bundling literature that each customer demands at most one unit of each product"*.
>
> **Ma & Simchi-Levi (2021, PBDC):** *''We study the monopolist pricing problem of a ﬁrm selling n diﬀerent items to a single random customer from the population. For each item, the customer wants at most one copy''*
>
> Despite this standard convention, our framework can easily accommodate scenarios where repeated products are allowed (up to a cap k).
>
> - **Transformation:** We can treat k copies of a single product type as k distinct "virtual products" (e.g., "Coke_1", "Coke_2"). This transforms a multi-unit problem into a standard distinct-item Mixed Bundling problem.
> - **Why Our Approach Suits This:** This transformation effectively increases the number of items from n to n×k. While this explosion in problem size ($2^n k$) would render traditional exact solvers intractable instantly, our GCN-based approach is specifically designed for scalability. As shown in our experiments, our model scales effectively to 100 products, meaning it is well-positioned to handle the increased dimensionality introduced by modeling repeated items as distinct virtual nodes.

---

### Official Review · Reviewer_uRPB · 2025-11-08

**Soundness:** 2
**Presentation:** 1
**Contribution:** 2
**Rating:** 2
**Confidence:** 4

**Summary:**

The paper proposes a GCN (graph convolutional network)-based approach to price bundles of products to customers. The network is firstly determined to output a small selection of bundles. Then, the selection of bundles is given to an MILP solver to determine the optimal prices on these bundles.

Baselines include an optimal but costly MILP approach and a fast but suboptimal Bundle-Size-Pricing (BSP) approach. Experiments show that the GCN-based approach preserves $>97.5\%$ optimal revenue and lower running time in small settings. In large settings, the GCN-based approach still outperforms BSP in terms of revenue and running time.

**Strengths:**

Learning to price bundles is an interesting idea. The neural network architecture is novel (as far as I can tell). The presentation is easy to understand. The experiments show the potential of strategic bundle-selection towards MILP-solving-based approaches to some extent.

**Weaknesses:**

The paper has some major weaknesses, any of which is a great challenge for the acceptance of this paper.

1) The paper does not explain how to train the GCN. In my understanding, the GCN is parameterized so training is a necessary step before using in experiments. In addition, the output of GCN is followed by an MILP solver, the latter of which is (probably) non-differentiable. Differentiating the target w.r.t. GCN parameters seems to be impossible.
2) The paper also does not clearly explain the experimental settings. For example, it is expected that the determination of all model parameters, such as $c _j^u, c _k^s, u _{kj}$, should be explained before experimental results.
3) The choice of baseline is questionable. The authors only select two baselines, only one of them (BSP) works in large settings. The authors have also mentioned that BSP is possibly suboptimal as it does not consider the heterogeneity of products. It seems likely that the authors should have considered more appropriate baselines as mentioned in Section 2, or at least given justifications on why other approaches are not appropriate baselines.

See below questions for my other concerns.

**Questions:**

1) The paper does not mention learning-based approaches on pricing bundles. Have the authors made literature review on this topic? Or is it the case that this topic remains blank in literature? (Although I believe the latter is quite unlikely.)
2) The GCN structure is designed with feature engineering without mathematical justifications. A simple MLP (with a post-processing sigmoid) may also work. Have the authors considered MLPs before GCN? Could the authors provide explanations on why GCN might be more appropriate than MLP? (Note that Ockham's Razor principle suggests that GCN is an overkill if it can be replaced with a simpler MLP without loss.)
3) The authors have emphasized that the $R _{kb}$ is non-additive in line 80-81. However, since there are only finite possible outcomes, why the non-additive utilities provide more difficulties than additive utilities? It seems to me that the additivity or non-additivity does not matter so much as the problem itself is finite.

---

> ### Author Response · Authors · 2025-12-02
> **choice of baselines**
>
> We thank the reviewer for identifying the need to better justify our choice of baselines. We acknowledge that the baseline selection appears limited; however, this is largely due to the specific constraints of our **Subadditive Valuations** setting, which renders many standard baselines inapplicable or theoretically incompatible.
>
> **1. Incompatibility of Additive Baselines (CPBSD, PBDC):** As noted in literature review, many pricing heuristics such as Component Pricing with a Bundle Size Discount (CPBSD) or Pure Bundling with Disposal for Cost (PBDC), rely on the additivity reservation and cost assumption (i.e., the value/cost of a bundle is the sum of the values/costs of its components). Our work focuses on subadditive reservation (representing substitution effects) and subadditive cost. We excluded them because they are theoretically misaligned with the valuation structure we are solving for.
>
> **2. Infeasibility of RL-based Methods:** While Reinforcement Learning (RL) is popular in dynamic pricing, it faces the curse of dimensionality in the context of combinatorial bundle generation. Therefore, current literature lacks RL-based solutions for the Bundle Pricing problem, particularly for scenarios requiring the simultaneous determination of bundle composition and price.
>
>
>
> **3. Bundle Recommendation:** While we acknowledge that both fields share the term "bundle," we argue that they address fundamentally different mathematical problems, making a direct baseline comparison infeasible.
>
> There is a significant difference in Objective Function. Bundle Recommendation aims to maximize prediction accuracy (e.g., Recall) by predicting which items a user prefers based on interaction history. In contrast, works on Bundle Pricing aim to maximize seller's profit subject to production costs and consumer surplus constraints. Joint bundle and pricing is much harder than bundle recommendation. A highly accurate recommendation can still yield poor profit if the pricing mechanism fails to capture consumer surplus. Besides, personal interaction history is not available for bundle pricing problems.
>
>
>
> **4. New Neural Baselines:** While Neural MILP, like Nerual Branching and Neural Diving, has shown great potentials in accelerating MILP solving process for a typical type of questions, it is not suitable for directly solving the MILP of mixed bundling even for problems with 10 customers and 15 products (1,343,498 decision variables). For either Neural diving or neural branching, the solver needs to put all decision variables into a huge graph, which makes it impossible to solve MILP under mixed bundling even for medium-size problems.

---

> ### Author Response · Authors · 2025-12-02
> **model training and experimental settings**
>
> We thank the reviewer for highlighting the need to clarify our training procedure and experimental settings. We agree that differentiating through a combinatorial MILP solver is generally infeasible. However, our proposed framework does not require differentiating through the solver. We include the training of our model and experimental settings in Appendix B. We train the GCN using 5000 samples with the optimal solution of problems (size m=10, n=10) given by mixed bundling.
>
> We treat the bundle selection as a link prediction task. The GCN is trained to predict the probability (score) of an edge existing between a product node and a segment node.
>
> - Labels: For the training set, we solve the instances to optimality using an exact solver (Gurobi) offline. If product i is assigned to segment j in the optimal solution, the label for edge (i,j) is 1; otherwise, it is 0.
> - Loss Function: We train the GCN parameters by minimizing the Binary Cross-Entropy (BCE) loss between the predicted edge scores and these pre-computed binary ground-truth labels.

---

> ### Author Response · Authors · 2025-12-02
>
> While learning-based approaches exist in the broader pricing literature, they primarily focus on **Personalized Pricing**, rather than the combinatorial Menu Design problem of Mixed Bundling.
>
> Works like Ettl et al. (2020) and Xue et al. (2016) use ML to tailor offers for individual users. This approach is one-to-one. However, the bundles and prices are very likely to be infeasible in the Menu Design framework since the utility maximization constraint may be violated. For bundle pricing problems, we have to make sure every customers are offered the same bundles choices.

---

> ### Author Response · Authors · 2025-12-02
> **GCN v.s. MLP**
>
> We agree that simpler models should be preferred when performance is equivalent. However, we did not choose GCNs for feature engineering purposes; rather, we chose them because MLPs are mathematically incapable of satisfying the core requirement of our framework: Size Generalization.
>
> We provide the mathematical and structural justifications below:
>
> - The Limitation of MLPs: A standard MLP requires a fixed input dimension. To use an MLP, we would have to fix the number of segments (m) and products (n) during training (e.g., m=10,n=10). Such a model cannot be applied to a test instance with m=20,n=15 without retraining (which degrades performance and efficiency).
>
> - The GCN Advantage: As shown in **Table 5.3** of our paper, we train our model on small instances and successfully apply it to instances with up to 100 products.
>
> - Mechanism: GCNs utilize **parameter sharing**. The learnable weights operate on the *feature dimension* of nodes, not the *number* of nodes. This allows the same trained GCN to process a graph of any size, which is impossible for a standard MLP.

---

> ### Author Response · Authors · 2025-12-02
> **subadditive valuations v.s. additive valuations**
>
> We thank the reviewer for this comment regarding the complexity of non-additive utilities. While we agree that the problem search space is technically finite (there are $2^n$ possible bundles), additivity is the deciding factor between a tractable problem and an intractable one.
>
> The mathematical justification for this difficulty gap is well-established in the classic literature, specifically Hanson & Martin (1990):
>
> 1. Additivity Allows Decomposition (Linear Complexity) As detailed in **Section 4 ("Pricing Large Numbers of Components") of Hanson & Martin (1990)**, when reservation prices and costs are **strictly additive**, the problem structure simplifies significantly.
>
> - Mechanism: Under additivity, the valuation of a bundle is simply the sum of its parts. This allows the problem to be decomposed or approximated using iterative methods that treat components individually.
> - Complexity: Hanson & Martin explicitly state that for the additive case, they provide a solution procedure that **"grows linearly with the number of components"**. This makes even large-scale problems not hard to solve using standard heuristics.
>
> 2. Non-Additivity Forces Enumeration (Exponential Complexity) When utilities are **non-additive** (e.g., sub-additive due to diminishing returns or super-additive due to complementarity), this decomposability property is lost.
>
> - Mechanism: The value of a bundle cannot be inferred from the sum of its components. The interactions between products mean that the pricing decision for a bundle of size k depends on the specific combination of items, not just their individual values.
>
> - Complexity: Consequently, the optimization cannot be simplified to component-level decisions. The solver must consider the full power set of  $2^n$ distinct bundles. As noted in the general formulation of Hanson & Martin (Eq 1-11), the number of constraints and variables "grows exponentially with the number of components."

---

### Meta-Review · Area_Chair_dNiA · 2026-01-05

**Summary:**

Reviewers' concerns are mainly around the following points:

W1. Unclear training protocol

W2. Unclear experimental setting

W3. Insufficient baseline

W4. Lacking theoretical insights

W5. Only use synthetic data, while industrial test is missing

W6. Scalability issue

W7. Unclear generalization performance

**Reviewer Concerns:**

Authors' responses did not fully address these concerns, though some clarification and explanation is helpful. In particular, authors did not respond to reviewer 6k7J's comments.

**Reviewer Scores:**

Given the incomplete response, it would be hard for the negative revewers to become positive.

---

### Decision · Program_Chairs · 2026-01-26

Reject